# Self- and Cross-Pollination in Argane Tree and their Implications on Breeding Programs

**DOI:** 10.3390/cells11050828

**Published:** 2022-02-27

**Authors:** Naima Ait Aabd, Abdelghani Tahiri, Redouan Qessaoui, Abdelaziz Mimouni, Rachid Bouharroud

**Affiliations:** Regional Center of Agronomic Research of Agadir, National Institute of Agronomic Research, P.O. Box 124, Avenue des FAR, Agadir 86350, Morocco; abdelghani.tahiri@inra.ma (A.T.); redouan.qessaoui@inra.ma (R.Q.); abdelaziz.mimouni@inra.ma (A.M.)

**Keywords:** *A. spinosa*, breeding programs, pollen, compatibility, self-pollination, cross-pollination

## Abstract

The argane tree (*Argania spinosa* L.) is a mostly self-incompatible species that must be cross-pollination. However, the cross-pollination is often insufficient to obtain a desirable fruit yield in the absence of compatibility between the orchard’s argane trees. Proper pollination design is therefore essential to ensure a supply of compatible pollen. In this study, pollen germination and pollen development following cross- and self-pollination were investigated in *A. spinosa*. The choice of compatible parents or a pollinizer is currently a new research topic for the production of argane fruits in the framework of argane farming programs. Different pollination experiments were designed with two main objectives: (i) to study cross/self-(in)compatibility in the argane tree, and (ii) to determine the degree of compatibility between selected superior genotypes for pollination strategies to improve fruit set in argane orchards. Thus, to determine if a pollination deficit exists, experiments were carried out on 14 genotypes, and 5421 flowers served as sampling. The germination rate of pollen was lower than 50% for three genotypes, and only four genotypes bloom twice a year. From cross-pollination trials, traits related to the mother trees, such as the shape of the fruit and fruit ripening duration, are not influenced by the gene flow transmitted by pollens. Self-pollination was very low (0.2%) for both hand- and free self-pollination but the highest fruit set rate observed was 5.3%. Based on the pollen effect study results, it can be concluded that different pollen sources affected the fruit set. Thus, the choice of an efficient pollinizer genotype must be inter-compatible with the main variety, bloom at the same time, and be regular (no alternation). This is the first time that a pollinizer tree was reported and studied for argane. To meet future argane farming requirements, the number and location of compatible pollinizers is very important in the argane orchard design. This design of pollination remains to be checked by alternately planting a row of pollinizer trees or inter-rows with main varieties.

## 1. Introduction

The argane tree (*Argania spinosa* L.) is an important forest species in Morocco and the only representative species of the Sapotaceae family. Its endemism accentuates the responsibility for the protection, rehabilitation, and conservation of this humanitarian heritage to benefit present and future generations. Recently, the United Nations General Assembly proclaimed 10 May the International Day of Argania. The argane tree is a xerophilic and thermophilic tree; it can only grow under specific temperature and humidity conditions [1]. The growth parameters as well as the bloom and fruit set depend on the rainfall between December and the end of March [2]. The fruit shape varies from fusiform, oval, apiculate oval, drop, rounded, globose, and a drupe containing a hard nucleus (fusiform, oval, or rounded) [3]. Furthermore, highly genetic diversity has recently been revealed and several argane populations have been studied based on several adaptive traits linked to the morphology and biometrics of trees, fruits, and seeds [3,4,5,6]. The huge diversity observed in the argane tree is be considered an advantage to implement a breeding program aiming for new productive cultivars and for domestication. Several research studies have focused on the assessments of trees in their environmental context. However, the argane tree is subject to natural selection, the impact of modern life and the increase in oil demands. Hence, understanding the mode of reproduction in *Argania spinosa* constitutes a necessary first stage for any strategy conservation, breeding, and selection for characters of interest in the argane tree. Several reproductive issues have been reported for this species. Some can be linked to the incompatibility observed between the tree’s genotypes. In fact, information on the compatibility system of *A. spinosa* is still unclear. Knowledge on the mode of pollination in argane remains very limited. Pollination is therefore an essential step in ensuring seed production. It is a critical stage in the sexual reproduction of plants [7]. In the wild, some argane trees have a strong flowering potential followed by a weak fruit set. In addition, the success of controlled cross-pollination can make possible the elucidation of reproduction systems in the argane tree. This work aims to study the argane reproductive system by studying the floral structure of argane, the viability and germination of pollens and the compatibility between trees genotypes to improve fruit yield. Thus, the identification of compatible genotypes can provide an important basis for the selection of the appropriate pollination trees for the modern farming system, “arganiculture”.

## 2. Materials and Methods

### 2.1. Plant Material and Site Study

The experiments were conducted in an *A. spinosa* orchard at the MelkZhar experimental farm of INRA in Belfaa, Agadir, Morocco (30.0434N; −9.55635W; 100m alt.), during 2018–2019 seasons. The climate is semi-arid with desert influences; and the average temperatures (max/min) are 24.8/11.5 °C, based on 10 years climate records. The average annual precipitation is 172.8 mm, with considerable variation from year to year. The orchard was planted in 2010 as an experiment of argane tree domestication at a density of 8 × 6 m. This orchard contains 142 elite trees selected from different locations of natural argane ecosystem.

### 2.2. Breeding System and Pollination Methods

To evaluate the effect of pollination methods on fruit quality, four treatments were considered: hand cross-pollination, hand self-pollination, autonomous self-pollination, and natural pollination as a control. Fourteen elite maternal parent trees were chosen from the 142 orchard trees. These 14 trees were split into two groups according to the bloom period. The first group includes eight trees and the second group six trees. Each tree plays both the role of donor and receiver of pollen in all possible combinations using a complete diallel analysis (Table 1 and Table 2). Therefore, the branches with flowers at the balloon stage were bagged to prevent any flower–pollinator interaction before starting the experiment. Thus, for each tree, seven branches were bagged for the first group and five branches for the second group. A total of 5421 flowers were served as sampling for hand cross-pollination and 2277 flowers for self-pollination or an autogamy test, including 1217 for hand self-pollination and 1060 for free self-pollination. The inflorescences used in these experiments were evenly and randomly selected. Four pollination method treatments were carried out to study the self- and cross-compatibility test: (i) natural or open pollination where floral buds were labeled to observe the rate of fruit-set under natural conditions, (ii) autonomous self-pollination where the flowers bagged were pollinated spontaneously with self-pollen, (iii) hand self-pollination where the flowers bagged were hand-pollinated with self-pollen and IV. Hand cross-pollination where controlled hand-pollinations were carried out in the morning between 07:30 and 10:30 a.m. (GMT+1) the day of anthesis. The pollen grains were dusted on the stigmatic surface of the emasculated flowers of the female parents when the stigma was receptive. The flowers were re-bagged and labeled. A second cross-pollination was conducted 24 h later. The inflorescences treated were then re-bagged with waterproof paper bags, and bags were not removed until fruit ripening time. Only, for self-pollination, the bags were removed one month after because all not pollinated inflorescences were fallen. The degree of crossability was determined as the percentage of fruit set one month after hand-pollination.

### 2.3. In Vitro Germination of Pollen

During the flower season (March–April 2018), branches with flowers at the mature stage were collected for each tree. Pollens were released from the anthers and sown in Petri dishes containing a solidified germination medium (1% agar, 20% sucrose, 200 ppm of CaCl2 and 75 ppm boric acid (H_3_BO_3_)). Cultures were incubated in the dark at 30 °C for 24 h [8]. A pollen grain was considered germinated if its pollen tube was longer than the length of the pollen grain. The observations were carried out under a light microscope (Optika DM-15, Italy) at magnification 100 X and photographed by an integrated digital camera. Pollen germination percentage was estimated as follows:$$\mathrm{Pollen}\;\mathrm{germination}\;\left(\%\right)=\frac{\mathrm{Number}\;\mathrm{of}\;\mathrm{germinated}\;\mathrm{pollen}\;\mathrm{grains}}{\mathrm{Total}\;\mathrm{Number}\;\mathrm{of}\;\mathrm{pollen}\;\mathrm{grains}}\times \;100$$

### 2.4. Flowering Observations and Fruit Maturity Cycle

The flowering period of parent’s trees was observed over the two consequent years (2018–2019) and the flowering stages were divided as follows: the beginning of bloom (first flower bud), full bloom (open flowers), and the end of bloom (total petals fall). The observations were conducted for each parent on three periods per year, from February to April (bloom once a year), from June to July (bloom twice a year), and over the year. Season phenograms of flowering were made for all evaluated trees based on the collected data of the bloom period. The fruit maturity cycle of each argane tree was monitored over two years (2018–2019).

### 2.5. Fruit Set

The fruit set was recorded 30 days after the end of bloom (at the initial fruit set). The fruit set per pollination treatment was calculated as the percentage of pollinated flowers developed to fruits:$$\mathrm{Fruit}\;\mathrm{set}\;\left(\%\right)=\frac{\mathrm{Number}\;\mathrm{of}\;\mathrm{fruits}}{\mathrm{Number}\;\mathrm{of}\;\mathrm{tested}\;\mathrm{flowers}}\times \;100$$

### 2.6. Statistical Analysis

Data were analyzed using a one or two-way ANOVA in Statistica software version 12. Significant differences between treatments at *p*<0.05 level were determined using the least significant differences (LSDs).

## 3. Results

### 3.1. In-Vitro Germination of Pollen

The germination percentage was significantly different depending on studied genotypes (Figure 1) based on observation in an agar-based medium composed of sucrose, boric acid (Figure 2). However, lower germination rates and shorter pollen tubes were observed among threedifferent parent flowers. Therefore, in vitro the germination percentage of pollen was found to be lower in three genotypes (INRA-94, INRA-67, and INRA-32) and the higher germination percentage of pollen was observed in 11 genotypes on the day of anthesis.

### 3.2. Blooming Periods, Floral Morphology and Fruit Maturity Cycle

Based on previous observations, in our domesticated orchard, *A. spinosa* trees bloom one to two times per year depending on genotypes. Therefore, the fruit bearing of some genotypes can start flowering at an early age, almost in the thirdyears. Indeed, full production was observed from 5 to 7 years. During flowering periods, all parent trees start to bloom in mid-February and for some parent trees a second flowering occurred during June–July (Table 3). In fact, based on our investigation, most trees (10 genotypes) have one flowering period per year (February-April) and four genotypes (INRA-135, INRA-132, INRA-98 and INRA-54) have two flowering periods per year (Table 3). The floral buds of argane tree, in particular, are hermaphrodite. For its development, the female organ reaches maturity earlier than the male organ. Therefore, the flower structure is bisexual and protogynous. Thus, the relationship between different anthers and stigmata position shows three morphs. The first is the mesostyle morph, where the position of anthers and stigmata are equal; the second is brevistyle, where the short style and long filament morph; and the third morph observed is longistyle, where the short filament and long styled morph. The floral morphs of 14 parent trees shown in Table 3, indicate the floral dimorphism existing in theargane species. The mesostyle morph was the dominant floral type (eigh tgenotypes) followed by brevistyle morph (five genotypes) and only one genotype had the longistylemorph.The fruiting heterogeneity was also observed during the fruit development (Figure 3). Thus, three distinct shapes of mature fruit were observed comprising the oval, drop, and round form (Table 3). Among the 14 genotypes studied, the oval shape is the dominant form and only one genotype has a rounded shape. The whole fruit maturation period for each genotype was established from the beginning of the fruit set to the end of fruit ripening (yellow color).

### 3.3. Cross/Self-(in)Compatibility Study and Pollination Strategies to Improve Fruit Set in Argane Orchards

#### 3.3.1. Cross-Compatibility

Fourteen domesticated genotypes of argane tree were used to initiate a complete diallel crosses program on two groups during the floral season March–April of 2018. The first group is composed of eight genotypes and six genotypes for the second group based on the flowering period. Almost 86 possible combinations of reciprocal crosses between these genotypes were made (Table 1 and Table 2). Almost all crosses made produced fruit indicating the presence of compatibility among genotypes, but the degree of this compatibility changes among genotypes. The inter-compatibility results during crosses between domesticated genotypes of *A. spinosa* are presented in Figure 3 and Figure 4. The inter-compatibility rate (expressed in %) often depended on the cross associations and varied from 0% to 55%. Sixty-four (64) combinations have a compatibility less than 30% and 12 combinations have a compatibility ranged from 30% to 55%; 10 combinations failed. Maximum fruiting was observed for parents INRA-67 followed by INRA-135, INRA-94, and INRA-132, when used as the female parent (pollen receptor). As a result, the highest percentage (50% and more) was observed in group 2 cross combinations (INRA-132 ♀ × INRA-94 ♂ (55%); INRA-94 ♀ × INRA-54 ♂ (54%) and INRA-67 ♀ × INRA-94 ♂ (50%)). Regarding group 1, only the combination INRA-135 ♀ × INRA-49 ♂ showed more than 50% (51%). Thus, the parents of group 2 have similar percentages suggesting a strong compatibility between the genotypes of this group. In addition, most studied genotypes (INRA-132, INRA-54, INRA-94) are compatibles with other parents and some (INRA-32, INRA-39, INRA-67) are incompatible (INRA-94). Therefore, cross-pollination between genotypes is essential for a good fruit set. The response of trees via the same pollen differs between *A. spinosa* genotypes. In some cases, there is a strong expression of the pollen flow depression via another mechanism included in Figure 4 and Figure 5. From these results, we can identify traits related to the mother trees, such as the shape of the fruit and the length of fruit ripening, which are not influenced by the gene flow transmitted by pollens. However, results based on the fruit set are difficult to interpret due to thelack of information on the mechanisms that control the fruit set in argane trees, even for hand-pollination. This original study highlights several problems that decrease the fruiting rate after intense flowering in argane trees.

#### 3.3.2. Self-Incompatibility

The results of hand- and free self-pollination are shown in Figure 6. In this study, self-pollination was very low (0.2%) for both hand- and free self-pollination. However, hand-controlled pollination presents a relative high fruit set rate of 5.3%. The fertilized flowers persist for almost a month (Figure 6) but after one month, there was a total drop infertilized flowers.

#### 3.3.3. Pollination Strategies to Improve Fruit Set in Orchards of Argane

Argane trees are self-incompatibles, they need cross-pollination between different varieties for fruits and a pollinizer plant that provides compatible pollen. A suggestion of future pollination design in argane orchards with pollinizers or parent compatibles is necessary as recommendations for successful argane fruit production. Therefore, the trees density and good placement of pollinizer or parent compatibles with respect to flowering periods (pollinizer and main variety bloom periods must overlap) are necessary to satisfy cross-pollination (Figure 7).

## 4. Discussion

Sexual reproduction is an essential process for generating genetic variation in the next generation. In this study, three types of pollination were carried out, cross and reciprocal pollination (diallel crossing), free self-pollination, and forced self-pollination. The objective was to determine the degree of inter-compatibility between argane genotypes and self-pollination. Thus, to identify some characters linked to the mother tree and those transmitted via the pollen gene flow.

Pollination and fertilization are the most sensitive phases of the annual cycle of the argane tree. The overlap of flowering periods, the compatibility between trees, the presence of pollinators or pollen vectors and good weather conditions influence their success. The orchard’s simultaneous flowering of trees provides sufficient pollen for pollination and fertilization, especially in trees with a short effective pollination period. In general, we found an overlap in all trees’ flowering times within the current study. Less synchronization was observed during the full flowering period between trees. It is noted that there is a potential risk of gene flow from male parent pollens tested as parents during controlled pollinations. The rejection and acceptance of pollen on the stigma depend on the genotype and the biochemical reactions initiated after pollen transfer [9]. Thus, the performance and success of fertilization depend on the interaction between the pollen tube and the pistil as well as on several environmental and endogenous factors [10,11,12,13,14].

For the rejection or acceptance of pollens, an understanding of pollen-style mechanical and cellular structures’ biochemical changes and interactions are required. Anthesis (dehiscence of anthers), viability, and pollen germination strongly depend on several factors (temperature, humidity, etc.). These pollen attributes play an important role in the crosses’ success [15]. Several studies report that anthesis occurs at night or early in the morning in monoembryonic cultivars, while anthesis occurs at night in polyembryonic cultures. Stigma receptivity can persist up to 72 h in some cultivars [15]. Tube growth, zygote formation, and fruit set are key aspects to consider in crossbreeding programs.

For all the studied argane trees, the fertilized flowers resulted in a much larger fruit set (83%) at the start, but the trees differed in the proportion of aborted fruits after pollination (the durability of the fruits). A decrease in the percentage of fruit from the controlled crossing over time was obtained in this orchard. At the beginning, there were 2421 manually fertilized fruits and only 285 seven-month-old fruits survived.

As a result, several factors or mechanisms can reduce the rate of incompatibility and allow the stigma to specify between the different pollen types it may receive. Some of them canbe related to environmental and endogenous factors [13,16,17,18,19,20]. However, until now, no scientific data havebeen reported on the argane tree.

Environmental factors include temperature, humidity, rainfall, and endogenous factors which include the age of the tree, position of inflorescences on the tree, floral structure (anther-stigma location), pollen-style interaction, pollen-egg fusion, and hormonal signs that occur during crossbreeding.

Cross-pollination in the argane tree can be performed by the wind or by pollinators as reported by several studies [8,21,22]. These studies confirm that insects play an essential role in the pollination of the argane tree. However, during our study of crosses in an argan orchard, insect’s and wind’s role as vectors of pollination arenot well-understood. During two months of work (coincides with the peaks of flowering and pollination) on the crosses in this argane orchard with spacing between the trees and rows was 6 m by 8 m, respectively, the involvement of insects compared withwind is very weak. According to the results obtained by Nerd et al. [21] in an argane tree orchard, the spread of its pollens by the wind is restricted to very short distances less than 6 m. Consequently, the wind has a strong intervention in disseminating pollens, especially when the trees are close to each other and provided that they have simultaneous flowering periods. At the same time, strong fruiting was observed in these trees compared with the previous year.

For self-incompatibility and self-compatibility, according to Ramirez and Brito [23], a tree is self-compatible if the Self-Compatibility Index (ISF) ≥ 0.30 and self-incompatible if the ISF<0.30. The indication of self-incompatibility in the argane tree was observed by Nerd et al. [21]. During pollen germination on the stigma, the tube reaches the base of the style within 72 h. This elongation of the pollen tube is similar for self and cross-pollination. However, the foreign pollen leads to a significantly higher fruit set (6.5%) compared with self-pollen (0.2%). The results obtained in this work (5.3% for cross-pollination and 0.2% for self-pollination) [21] agree with those obtained in a domesticated argane orchard.

The exploitation of knowledge on self-incompatibility mechanisms in flowering plants is very useful. Since Darwin’s [7] studies, considerable knowledge has been acquired about these mechanisms of self-incompatibility. Recently, for other species, incompatibility systems create barriers to avoid self-fertilization and promote cross-pollination [17,19,24,25,26,27,28,29]. Self-incompatibility (SI) is genetically controlled, especially during pollen/stigma recognition. It is classified into two systems: gametophytic auto-incompatibility, which is determined by the haploid genotype of pollen; and sporophytic auto-incompatibility which recognizes the diploid genotype of the mother plant providing the pollen.Recently, a third system (pre- and post-zygotic) was reported. It is characterized by reaching the ovary and triggering mechanisms that inhibit embryo development.

For the argane tree, the blocking does not occur at the level of the pollen-stigma interaction since the self-pollen succeeds in germinating and reaching along with the style [21] (Figure 8). It is concluded that some degree of postzygotic differentiation results in the inability to produce zygotes after self-pollination.

## 5. Conclusions

The variation in crossbreeding ability between argane genotypes in the present study revealed that parental selection affects sexual compatibility. Thus, the degree of compatibility between the parents results in a great genetic variability that can be promising for future genetic improvement of the argane tree concerning desirable horticultural traits. Flowering timing is a major factor in the hybridization potential in the argane tree, as three flowering periods differ from genotype to genotype in the orchard. Therefore, the availability of genotypes with high flowering and compatibility is very demanding to consider during orchard installation for argane cultivation, taking into account favorable climatic conditions and vectors for good pollination.

## Figures and Tables

**Figure 1 cells-11-00828-f001:**
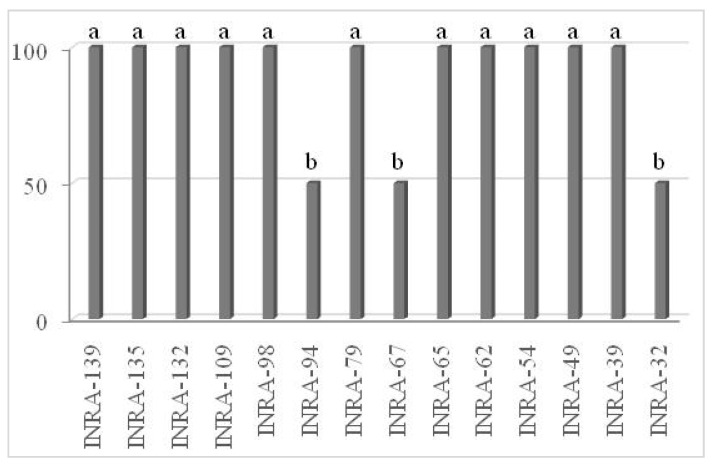
In vitro pollen germination percentage of different cross-pollinated genotypes indicate significant differences at *p* ≤ 0.05 by the LSD test, a,b letters indicate homogeneous groups.

**Figure 2 cells-11-00828-f002:**
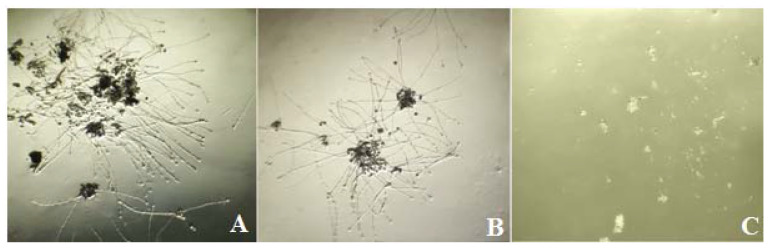
In vitro pollen germination test on a germination medium (1% agar and 20% sucrose), (**A**) high pollen germination with tube growth, (**B**) moderate germination of the pollen tube, (**C**) no germination of pollen tube.

**Figure 3 cells-11-00828-f003:**
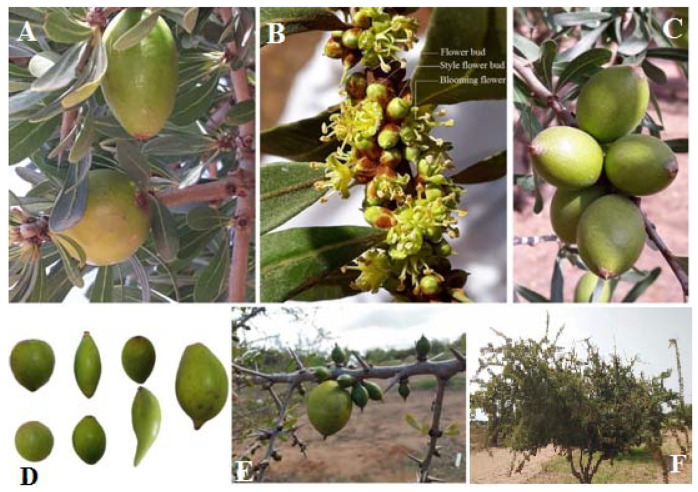
Blooming and fructification of argane tree. (**A**) Fruit variability within the tree; (**B**) blooming phenology; (**C**) fruit production; (**D**) phenotypic diversity of fruit in the orchard; (**E**) different stages of the fruit of the tree with two-flowering period/year; (**F**)argane pollinizer tree.

**Figure 4 cells-11-00828-f004:**
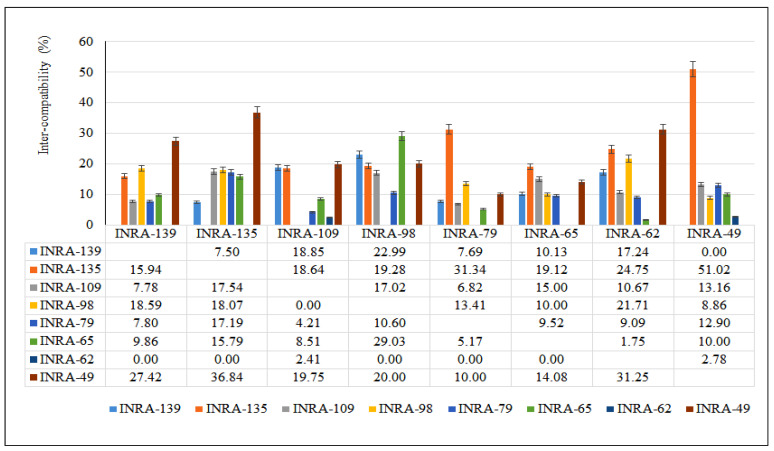
Compatibility indices of *A. spinosa* using cross-diallel programs (8 × 8): each female parent crossed with seven male parents.

**Figure 5 cells-11-00828-f005:**
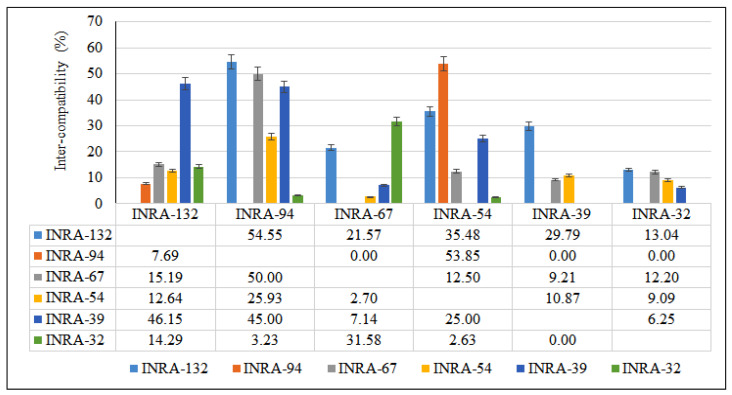
Compatibility indices of *A. spinosa* using cross-diallel programs (6 × 6): each parent crossed with five male parents.

**Figure 6 cells-11-00828-f006:**
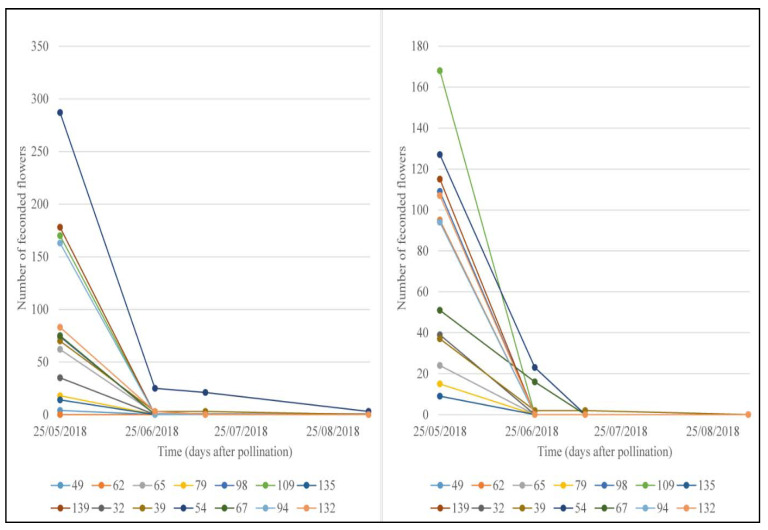
Evolution of the number of fertilized flowers after hand (**left**) and free self-pollination (**right**).

**Figure 7 cells-11-00828-f007:**
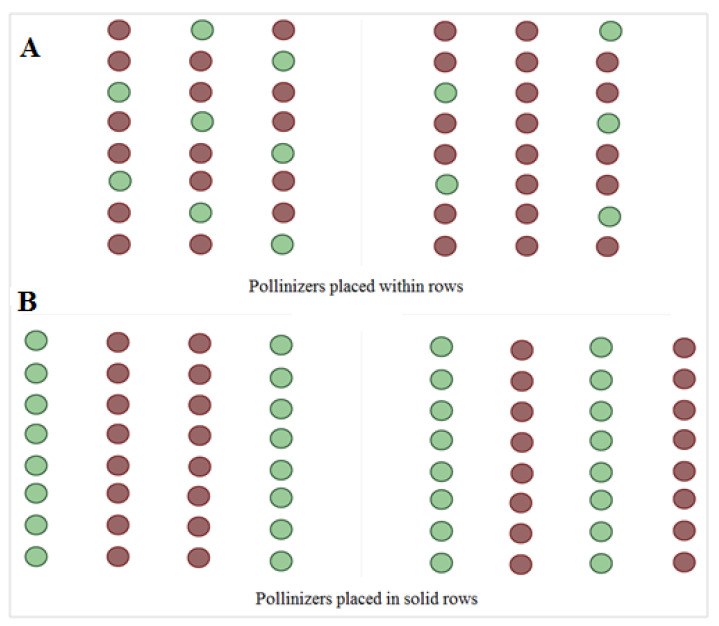
Arganepollination management and orchard design. Intermixed design (**A**) or alternative full row method (**B**). Pollinizers are represented by the green circle.

**Figure 8 cells-11-00828-f008:**
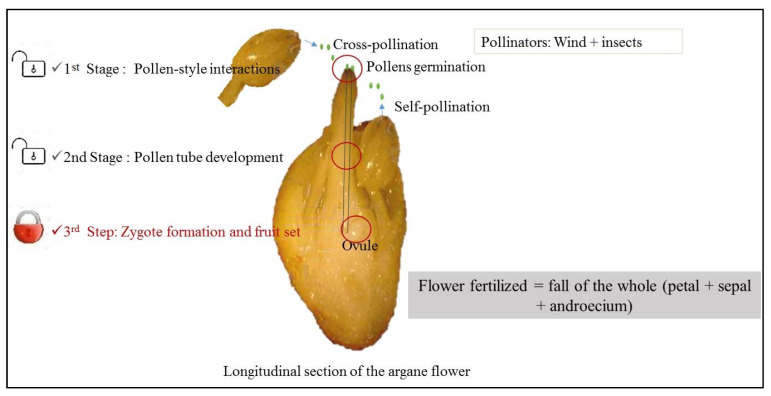
Illustration for self-fertilization blocking levels and hypothesis for the limitation of zygotes development.

**Table 1 cells-11-00828-t001:** Completediallel program for the 1st group (8 × 8).

		INRA-139	INRA-135	INRA-109	INRA-98	INRA-79	INRA-65	INRA-62	INRA-49
♀♂									
INRA-139			√	√	√	√	√	√	√
INRA-135		√		√	√	√	√	√	√
INRA-109		√	√		√	√	√	√	√
INRA-98		√	√	√		√	√	√	√
INRA-79		√	√	√	√		√	√	√
INRA-65		√	√	√	√	√		√	√
INRA-62		√	√	√	√	√	√		√
INRA-49		√	√	√	√	√	√	√	

**Table 2 cells-11-00828-t002:** Complete diallel program for the 2nd group (6 × 6).

		INRA-132	INRA-94	INRA-67	INRA-54	INRA-39	INRA-32
♀♂							
INRA-132			√	√	√	√	√
INRA-94		√		√	√	√	√
INRA-67		√	√		√	√	√
INRA-54		√	√	√		√	√
INRA-39		√	√	√	√		√
INRA-32		√	√	√	√	√	

**Table 3 cells-11-00828-t003:** Flowering and fructification observations of 14 genotypes cross-pollinated.

Genotypes	Fruit Maturity Cycle	Fruit Shape	Floral Morphs	Number Of Flowering Periods/Year
INRA-139	12 months	Round	Mst	1 time/year (March–April)
INRA-135	9 months	Round	Bst	2 times/year (March–April) and (June–July)
INRA-132	12 months	Drop	Bst	2 times/year (March–April) and (June–July)
INRA-109	12 months	Oval	Bst	1 time/year (March–April)
INRA-98	12 months	Oval	Bst	2 times/year (March–April) and (June–July)
INRA-94	12 months	Oval	Mst	1 time/year (March–April)
INRA-79	12 months	Oval	Mst	1 time/year (March–April)
INRA-67	12 months	Oval	Lst	1 time/year (March–April)
INRA-65	15 months	Oval	Mst	1 time/year (March–April)
INRA-62	12 months	Oval	Mst	1 time/year (March–April)
INRA-54	12 months	Oval	Bst	2 times/year (March–April) and (June–July)
INRA-49	12 months	Oval	Mst	1 time/year (March–April)
INRA-39	12 months	Oval	Mst	1 time/year (March–April)
INRA-32	12 months	Oval	Mst	1 time/year (March–April)

Mst: mesostyle; Bst: brevistyle; Lst: longistyle.

## Data Availability

All data generated in this work are presented within the manuscript.

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
