# Peer review of "Self- and Cross-Pollination in Argane Tree and their Implications on Breeding Programs"

_cells, 2022, doi:10.3390/cells11050828_

Round 1
Reviewer 1 Report
The manuscript entitled “Self and cross pollination in argane tree and their implications on breeding programs” is a well described study of pollination experiments for argane trees. I would accept it for publication after minor revision points:
- Line 34: change “sapotaceae” to “Sapotaceae”
- The species name A. spinosa has to be change to italic letters in the complete text
- Please control the space characters in the complete text
- Line 133-134: Rephrase the incomplete sentence: “in an agar-based medium composed of sucrose, boric acid”
- Line 136: Specifiy “lower germination percentage”. Is the germination rate really 50% in the three genotypes, and all other comprises 100% as shown in Figure 1. How many pollen tube were counted per genotype? Specifiy the detailed numbers in Materials and methods part or in Figure 1. If statistical data are presented (indicated by a- and b-letters in Figure 1), the authors have to present detailed numbers or standard deviations in Figure 1.
- Rephrase the incomplete sentences in Line 194-196
- Line 210: change “self-incompatibility” to “self-pollination”
- Line 210-212: It is not clear how the self-pollination rate of 0.2% and the fruit set rate 5.3% is calculated based on the data presented in Figure 5. Please present more explanation and detailed data regarding that point. Specify the word “feconded flowers” in Figure 5.
- Line 295: change (IS) to (SI) and delete “a” in this sentence.
Author Response
Dear Reviewer
Please see the attachment
Regards

Reviewer 2 Report
While your results are of potential interest (first time that pollinizer tree was used and studied for argane tree), the topic of your manuscript falls outside of the scope and level of this journal. For an overview of the Aims & Scope, please have a look at the journals’ homepage. In addition to viability and pollen germination and compatibility studies, it is necessary to study other key factors relevant to pollination and fertilization that affect yield. Much has been written about these factors in discussion section.
We hope you will consider the journal for publication of future studies within the scope. For alternative journals that may be more suitable for your manuscript, please refer to MDPI site.
Author Response

(The authors gave the same response as above.)

Reviewer 3 Report
In the study, the authors investigated the self and cross pollination on the fruit yield of argane tree. They found some important findings, such as fruit set rate is only 0.2% for both hand and free self-pollination while it is 5.3% for hand-controlled pollination. Thus, the results will be valuable for fruit production of argane tree. I have several minor issues.
- It will be better to supplement some representative images for pollen germination of different cross-pollinated genotypes.
- The authors are suggested to add some images related to pollen germination on the pistil for different pollination methods and genotypes.
- The conclusion should be shortened.
Author Response

(The authors gave the same response as above.)

Round 2
Reviewer 2 Report
The name of the special issues where you have submitted this manuscript is "Omics in Plant Genetics and Breeding". According to the vocabulary OMICS aims at the collective characterization and quantification of pools of biological molecules that translate into the structure, function, and dynamics of an organisms. The manuscript that is submitted in 'Cells' ('Self and cross pollination in argane tree and their implications on breeding programs') does not cover any structural, functional, or dynamics study in reproductive biology of the argane tree. It covers only pollen germination, fruit set and pollination strategies to improve fruit set in orchards of argane (and that is spatial distribution of trees within the orchard). Described methods and results are far away from enough to fit into the special edition.
The manuscript is not suitable for the 'Cell' manuscript, and should be rejected.